# Psychological Resilience May Be Related to Students’ Responses to Victims of School Bullying: A Cross-Sectional Study of Chinese Grade 3–5 Primary School Students

**DOI:** 10.3390/ijerph192316121

**Published:** 2022-12-02

**Authors:** Lu Lu, Liping Fei, Yunli Ye, Maoxu Liao, Yuhong Chang, Yiting Chen, Yanli Zou, Xin Li, Rong Zhang

**Affiliations:** 1School of Public Health, Southwest Medical University, Luzhou 646000, China; 2National Center for AIDS/STD Control and Prevention, Chinese Center for Disease Control and Prevention, Beijing 100000, China; 3Information and Education Technology Center, Southwest Medical University, Luzhou 646000, China; 4Luzhou Center for Disease Control and Prevention, Luzhou 646000, China

**Keywords:** school bullying, psychological resilience, response, primary school students, cross-sectional study

## Abstract

Background: School bullying is a major public health problem with a large impact on children’s health. There is an increasing number of cases of school bullying reported in China. Studies have shown that the health consequences of different ways of responding to school bullying may be quite different and that psychological resilience is also closely related to aggressive behaviors. However, there has been little research on whether individuals with different psychological resilience levels respond differently. Objective: To explore the relationship between responses to school bullying and psychological resilience, which may provide new ideas and strategies to better prevent and intervene in school bullying. Methods: A random sample of 5425 primary school students aged 7–14 years were recruited in Luzhou, China. All students completed a questionnaire anonymously. The statistical significance of differences between groups was tested using the *χ*^2^ test or t test. Binary logistic regression was conducted to explore the relationship between responses and psychological resilience. Results: Over two-thirds of primary school students in this study reported experiencing bullying in the past year. The rate of positive responses among victims was 69.10% (2596/3757, 95% *CI*: 67.62~70.58%). There was a positive relationship between psychological resilience and positive responses. This relationship was observed for all victims (*OR* = 1.605, 95% *CI*: 1.254~2.055), especially male victims (*OR* = 2.300, 95% *CI*: 1.624~3.259). Conclusions: There was a positive relationship between primary school bullying responses and psychological resilience among victims, with differences by sex. Therefore, increasing students’ level of psychological resilience, possibly by improving their responses, is important for preventing school bullying. Meanwhile, effective interventions for school bullying should be developed from multiple perspectives, particularly sex, bullying roles, and psychological resilience.

## 1. Introduction

School bullying is a global public health problem and a serious social issue; it refers to the behavior that occurs between students, one individual deliberately or maliciously bullies and insults another through physical, verbal or network means, causing personal injury, property damage, or mental damage [1]. School bullying that occurs among children and adolescents has become one of the most serious public health threats and causes social, economic, and medical effects that urgently need to be addressed. According to UNESCO statistics for 144 countries in 2018, approximately one in three children suffered from school bullying. However, the incidence of school bullying varies by region, ranging from 7.1% to 74.0% [2]. One in two children in the world was found to have experienced school bullying [3]. A previous study showed that nearly a quarter of school-age children in the United States had experienced school bullying during the previous school year [4]. In India, approximately 60% of school students aged 8–14 years were reported to be involved in school bullying [5]. Compared with other countries, Chinese research started relatively late. It was not until November 2017 that central national bodies provided a clear definition of school bullying at the national policy level in the “Integrated Governance Program for Strengthening School Bullying Management among Primary and Middle School Students”. However, school bullying was still prevalent among Chinese students. A study showed that 33.36% of primary and secondary school students suffered school bullying in 29 counties in China in 2016 [6]. In 2019, researchers studying the prevalence of school bullying, its causes, and severity found that 66% of students in Mainland China reported bullying incidents [7]. Numerous studies have shown that school bullying is influenced by various factors that continuously and simultaneously interact, including the individual, family, school, community environment, etc. [8,9,10]. School bullying poses severe and profound health hazards. Suffering or witnessing school bullying over a long period of time may lead to victims’ death or physical disability, may distract students from learning, may cause students to be truant or even drop out of school altogether; further, these experiences may lead students to experience a series of psychological and social issues, such as low self-esteem, depression, and suicidal ideation [11]. It may even lead to problems during adulthood [12].

Although school bullying can result in numerous adverse health consequences, the same bullying behavior has very different consequences for different individuals [13]. Different individuals’ responses to the same bullying behavior can also be very different. For example, some individuals choose to be silent or leave, while others choose to ask for help or resist. Coupled with the fact that different responses to the same bullying behavior may have dramatically different health consequences and effects on the repetition and continuity of bullying behavior, which may be related to the subsequent development of both the bullies and the victims. Some studies suggested that psychological resilience plays an important protective role in problem behaviors and aggressive behaviors [14,15] and in the positive response to poor childhood experiences [16]. However, the relationship between responses and psychological resilience in school bullying is less well documented.

Our study applied the resilience development model of ecological theory [17] to examine the effect of resilience on the response of primary school students. Resilience is the process by which an individual interacts with the internal and external environment when encountering various traumas and blows and eventually obtains positive adaptive consequences [18]. Response refers to the cognitive and behavioral efforts that individuals make to balance their mental state when facing a stressful environment or suffering a stressful event [19] and include positive responses (characterized by aggressiveness, initiative, activeness, and outwardness) and negative responses (characterized by avoidance, passivity, inhibition, and inwardness) [20]. Understanding response and psychological resilience together, it can be argued that the response chosen by individuals is an expression of their psychological resilience when controlling adversity or stress [21], and it can also be argued that response is an important window of expression of individuals’ psychological resilience when dealing with stress [22], so we believe that there is a relationship between response and resilience in school bullying. At the same time, it has been shown that the level of psychological resilience is closely related to responses, with individuals with high psychological resilience tending to use more proactive strategies [23]. Therefore, this study will be significant in exploring the relationship between different levels of psychological resilience and responses to the prevention and control of school bullying.

Recent research on school bullying has mainly focused on determinants and health effects. The largest cohort studied was composed of college students, followed by middle school students. However, research has shown that most respondents have experienced and encountered school bullying since primary school [24,25], which is an important stage in an individual’s physical and mental development and behavioral habit formation. In addition, school bullying is a long-term, potentially influential process. The younger an individual is, the greater the impact of school bullying [25,26]. However, primary school students, as a population at high risk of engaging in school bullying, have rarely been included in studies. Previous studies have reported differences in psychological resilience between males and females [27]. However, few studies have examined the impact of psychological resilience on students’ response to school bullying and its sex differences. This study collected information on the psychological resilience levels and the response of the bullied through a cross-sectional study to analyze and validate the relationship between response and psychological resilience levels and provide more insights into the correlations of responses to school bullying and behavioral outcomes. The present study could support further efforts for the intervention and prevention of school bullying based on the following three major research objectives: (a) to understand the status of responses and psychological resilience among primary school students; (b) to explore the determinants of students’ responses to school bullying; (c) to determine whether psychological resilience is associated with responses to school bullying and the strength of the association.

## 2. Materials and Methods

### 2.1. Study Design and Sample

This cross-sectional study was conducted by the research team of Southwest Medical University in Luzhou, Sichuan Province. The data for this article were collected from 6066 students from grades 3~5 of 12 different primary schools in the urban area of Luzhou. In this study, 5425 students were included in the analysis. Luzhou city is a third-tier city in China located in the south-central part of Sichuan Province. Its economic development level is representative of the vast majority of cities in Central China.

Multistage random sampling was used in the study. A total of 144 primary schools in three urban areas (Jiangyang District, Longmatan District, and Naxi District) of Luzhou city were divided into three grades (high, medium, and low) according to their economic status. With reference to the yearbook of primary students’ statistics, the proportion of public and private primary school students across the three different economic categories was 5:1. Therefore, in the first stage, three schools in each division, including one private school and two public schools, were randomly selected. In the second stage, all classes in grade 3~5 were selected.

Data were collected with a self-designed questionnaire, which was revised on the basis of a large number of previous studies and through an investigation. The questionnaire, which has been published elsewhere [28,29], was adapted by the research team for the survey. A self-completed questionnaire was administered by investigators, who underwent standardized training from October 2018 to January 2019.

### 2.2. Measures

The measurement tool used in this study was the questionnaire, which mainly collected data on participants’ sociodemographic information, school bullying, and psychological resilience.

#### 2.2.1. Sociodemographic Variables

The sociodemographic variables of the participants included age, sex, grade, character (quiet vs. general vs. outgoing), and the enrolled schools (public or private). Public schools are funded by the government, and staff is appointed by the education department. Private schools refer to schools funded by an individual or a private institution and approved by the local government and education department. In addition, other background information included the number of good friends (≤1 vs. ≥2); class cadre participation (yes vs. no); academic performance (average vs. above average vs. below average); relationship between siblings (good vs. general vs. poor); one-child family (yes vs. no); engagement in bad behavior, such as smoking, playing computer games, dropping out, and wandering and not going home after school (yes vs. no); sufficient sleeping time ≥8 h (yes vs. no); education level of parents (university and above vs. high school vs. junior high school vs. elementary school and below); parents as the main family educator (yes vs. no); parents quarrel in front of the children (yes vs. no); divorced parents (yes vs. no); method of education after making mistakes (negative education vs. positive education vs. both).

#### 2.2.2. School Bullying

The measurement of school bullying was assessed by a bully/victim questionnaire for primary students revised by Professor Zhang Wenxin [30]. The revised questionnaire was based on the Olweus Bully/Victim Questionnaire for primary students [31] and was suitable for the situation in China. The revised questionnaire contains two sections, bullying and being bullied. Each section has seven questions, which ask respondents how many times they have bullied others or been bullied in the past year. Each question has five options, 0 times, 1–2 times, 3–4 times, and 5 times or more, which are scored as 0, 1, 2, and 3 points, respectively. If respondents bullied others or were bullied more than once in the past six months, they were identified as bullies or victims, respectively. The Cronbach’s α for the school bullying scale for the pilot sample was 0.77, and the KMO statistic was 0.76.

#### 2.2.3. Respondents’ Responses to School Bullying

The measurement of responses adopted scholars’ classification standard [20,32], which is widely recognized in academic circles in China. According to this classification standard, the paper adopted a self-designed questionnaire that is suitable for Chinese primary school students to assess responses to school bullying. The questionnaire included one question on the positive and negative responses of victims: “What did you usually do after being bullied? (You can choose more than one)”. This question has six options: 1 = “Pretended it never happened”, 2 = “Told the teachers”, 3 = “Told my parents”, 4 = “Fought back”, 5 = “Asked someone else to fight back”, 6 = “Told a friend”. Among these responses, 2, 3, and 6 are positive options, and 1, 4, and 5 are negative options. Respondents were defined as having a positive response when they selected at least one positive option and no negative option (i.e., they selected “2”/“3”/“6”/“2&3”/“2&6”/“3&6”/“2&3&6”); respondents were defined as having a negative response when they selected at least one negative option (the answer had to contain one of the following: “1”/“4”/“5”/“1&4”/“1& 5”/“4&5”/“1&4&5”).

#### 2.2.4. Psychological Resilience

The measurement of psychological resilience was evaluated by using the Ego Resiliency Scale (ER89) developed by Block and Kreman (1996). The English version of this scale has been translated, retranslated, culturally adapted, assessed by relevant experts, and widely used to measure resilience in China [33,34]. There are 14 items in the scale scored on a 4-point scale: 1 = “Does not apply at all”, 2 = “Applies slightly, if at all”, 3 = “Applies somewhat”, and 4 = “Applies very strongly”. The scale has a total score of 56 points. The higher the score is, the better the psychological resilience, and the easier it is to recover when encountering stressful events. In the pilot sample, Cronbach’s α for the psychological resilience scale was 0.76, and the Kaiser–Meyer–Olkin (KMO) statistic was 0.87. In this study, psychological resilience was divided into three levels based on participants’ scores by the tertile method: a lower level, with 37 points and less; an average level, with 38~44 points; and a higher level, with 45 points and more.

### 2.3. Analysis

Epidata software (version 3.1) was used to establish a database, and a double-entry method was adopted to ensure the quality of data entry. Statistical analysis was performed using the IBM SPSS Statistics (version 26). Microsoft Excel (version 2019) was used as a drawing tool. Demographic characteristics and other information were descriptively analyzed. Numerical data are shown as the means ± standard deviations, and count data are displayed as proportions or ratios. The *χ*^2^ test for examining the relationship between the positive response of victims and qualitative variables (such as sex, grade, character, number of good friends, etc.), and *t* test for examining the relation between positive response and quantitative variables (such as age, recognition score, etc.). Whether the victim’s response to school bullying was positive or not was used as the dependent variable (yes = 0, no = 1), and factors that were significant and professionally considered significant in the univariate analysis were used as the predictors, where multi categorical variables such as psychological resilience level were dummied, and binary logistic regression was conducted to explore the relationship between responses and psychological resilience. *p* values were 2-sided, and statistical significance was set at less than 0.05.

### 2.4. Ethical Consideration

Written consent was sought from the school management, and informed consent was obtained from the caregivers of all participants. The study was approved by the Ethics Committee of The Affiliated Hospital of Southwest Medical University (No. KY2019128).

## 3. Results

### 3.1. Description of the Sociodemographic Characteristics of the Participants

A total of 6066 students were surveyed in the study, and 5425 responses were valid. The sociodemographic characteristics of the respondents are shown in Table 1. The average age of the students was 9.53 ± 0.97 years, and more than half (52.15%) were male. Approximately 90% of students had good relationship with their siblings. Nearly half of the children (56.82%) were educated in positive ways after they made a mistake. The mean psychological resilience score was 40.52 ± 8.04, which was at the middle and lower levels (Table 1).

### 3.2. Participants’ Experience and Responses to School Bullying

In the survey of school bullying incidents that occurred in the past year, the prevalence of bullying, bully–victim, and victim in primary schools was 71.71% (3890/5425), 27.80% (1508/5425), and 69.25% (3757/5425), respectively, and nearly 70% of victims reported that they chose to respond positively by telling the teachers (54.62%), telling their parents (48.71%) or telling a friend (17.43%) (Figure 1).

### 3.3. Response to School Bulling among Victims with Different Characteristics

The results showed that factors affecting the responses of victims including district, sex, grade, character, number of good friends, academic performance, relationship with siblings, bad health behavior, sufficient sleeping time, mother’s education level, parents as the main family educators, parents quarreling in front of the children, divorced parents, education method, age, recognition of school bullying score and psychological resilience. These factors differ significantly between students with positive and negative responses (*p* < 0.05) (Table 2).

### 3.4. Logistic Regression Analysis of Factors Influencing Victims’ Response to School Bullying

Table 3 shows that there was a positive relationship between psychological resilience and positive response to school bullying: the better the psychological resilience was, the higher the positive response rates. This relationship was observed for all victims (*OR* = 1.605, 95% *CI*: 1.254~2.055) and particularly male victims (*OR* = 2.300, 95% *CI*: 1.624~3.259). In addition, the mother’s high level of education was related to male victims’ positive responses, while female victims’ positive responses were related to an outgoing character and parents as the main family educators. Among victims, those who had good relationships with their siblings had sufficient sleeping time and were educated in positive ways after they made a mistake and also chose to respond positively. Male victims were more likely to choose negative responses than female victims (Table 3).

## 4. Discussion

Responses to school bullying are strongly linked to health outcomes [35], while responses may also be related to resilience. The primary school period is a crucial period for the formation of adolescents’ thoughts and behaviors. Students’ responses during this life stage influence the repetition and health outcomes of school bullying [12]. It is important to understand psychological resilience and responses to school bullying and to explore the relationship between behavioral outcomes to intervene and prevent school bullying. This study found a positive relationship between psychological resilience and the responses to school bullying among primary school students in Luzhou, and there were sex differences in victims of school bullying.

Our findings suggested that the responses of victims were relatively positive. A total of 69.10% of the victims chose positive responses, mainly seeking help from teachers and parents, which was higher than the proportion of Haerbin primary school students [36]. The reasons may be as follows. First, the contents of the questionnaire used in this study were different from that of previous studies. Second, the study participants were different. There are differences in the economic and educational levels of the regions where the participants of different studies were located. Some studies have pointed out that various economic and educational levels are closely related to school bullying [37]. Alternatively, the positive response rate may indicate that the status of bullying response among primary school students is improving in China, due to the influence of various online media and higher levels of attention to school bullying from families and schools. Although the positive response to primary school bullying in this survey was relatively high, we have to consider those students who did not respond positively. Nearly one-third of the victims chose to respond negatively, which indicates that it is imperative to develop interventions for preventing and reducing the occurrence of school bullying.

The psychological resilience of pupils in the present study was at the middle and lower levels (average score of 40.00 ± 8.09), which was lower than similar studies conducted in China [38]. Psychological resilience can be a protective factor for mental health outcomes over and above adverse childhood experiences [16] and can enable people to use various resources to deal positively with crises or stress [39]. The definition of psychological resilience emphasizes the dynamic and changing nature of individual psychological resilience, and the interaction between psychological resilience and the environment will promote the development of individual psychological resilience [18]. The resilience development model [17] also considers psychological resilience as a result of a successful response, which is essentially the process of response to various stress scenarios throughout the life cycle. It may be that the results of the response lead to the accumulation of psychological resilience in the average person’s life and that the accumulating positive consequences shape the psychological resilience of the individual. Primary school students are at a critical stage of physical and mental development and growth, and their mental toughness is still immature and highly moldable. Therefore, parents, schools, and society may focus on cultivating and promoting the positive development of children’s psychological resilience.

Similar to other studies [40,41,42,43,44], we found that many factors were related to the choice of response, such as the relationship with siblings, sleeping time, method of education after making a mistake, character, number of good friends, mother’s education level and parents as the main family educators. After controlling for confounding factors, we found that individuals with high psychological resilience were more likely to choose positive responses, which was consistent with the results of previous research [45,46,47]. Individuals with high levels of psychological resilience chose positive responses to reduce the impact of negative events in life, indicating that psychological resilience may buffer adverse effects by influencing choices of individual behavior, which was in agreement with the opinion of Faria et al. (2014) [39]. Rutter sees psychological resilience as a relatively stable psychological quality or ability similar to a personality trait [21]. Individuals with good psychological resilience characteristics are better able to respond and recover in the face of trauma and shock [48]. Under stress conditions, psychological resilience exists more as a direct antagonist that can mitigate the adverse effects of risk factors [49]. Meanwhile, Budge’s study [50] shows that when individuals tend to adopt positive responses, their anxiety and depression levels also decrease, and positive responses are protective factors for the mental health of victims of school bullying. Some researchers also believe that positive responses not only allow victims to face various negative events better but also enhance their self-esteem to improve their life adaptability and overall physical and mental health development [51]. Maslow’s Hierarchy of Needs [52] argues that safety is the most basic human need and that gaining peer respect and improving self-efficacy through positive responses to school bullying is a need for self-actualization and an indispensable prerequisite for children’s healthy physical and psychological development. It is evident that improving the victim’s response to bullying by increasing the level of resilience may be important for preventing and intervening in school bullying. Therefore, parents and teachers should nurture children’s psychological resilience from an early age, so that children learn to have courage and to face and solve difficulties and challenges from a more positive perspective.

Moreover, we found that there were significant sex differences in the relationship between psychological resilience and responses. The results of the logistic regression model showed that only male victims’ psychological resilience was related to responses. This result may be related to the sex role requirements that Chinese parents instill in their children. In China, most parents expect boys to be strong, brave, independent, and dare to face difficulties [53]. Therefore, when the ability of the victim is not much different from that of the bully, boys may react instinctively and use violence to control violence; when the ability of the victim is far weaker than that of the bully, out of vanity and fear, male victims are more likely to choose forbearance and silence. Under such circumstances, boys with poor psychological resilience are more likely to respond negatively. This is consistent with the psychological compensation model [54] that explains psychological resilience as a factor that directly counteracts stress or risk factors. Boys with high psychological resilience are already able to spare themselves from the negative effects of adversity without having to mobilize their cognitive coping capacity, whereas boys with low psychological resilience may be unable to mobilize resources and function as coping as they should because of their low psychological resilience. As a result, men with poor psychological resilience are more likely to choose to respond negatively. Alternatively, it may be due to male and female character traits. Men are more impulsive than women, so resilience may have a stronger buffering effect on male behavior.

This study has several limitations. First, considering the nature of the cross-sectional design, the causality between psychological resilience and responses to school bullying could not be established in this study. Longitudinal studies should be used to further understand causality in future research. Second, the subjects of this study were a large sample of primary school students from Luzhou city, Sichuan Province, which is a typical third-tier city in China. In view of the different socioeconomic conditions between cities, the generalizability of the findings to populations in other cities, especially those with other socioeconomic conditions, is limited. Third, given that school bullying is a sensitive issue, all the measures in this study were self-reported. Although the purpose and significance of this research were explained to the participants before the survey and the questionnaires were completed independently and anonymously to minimize bias, there may still be a gap between the students’ reports and reality. More empirical studies are needed to replicate our findings. Fourth, this research only analyzed the differences between victims in the relationship between primary school bullying responses and psychological resilience. However, bullying roles also include perpetrators, bystanders, and agitators. In the future, we can further analyze the relationship of other bullying roles.

## 5. Conclusions

Using population-based data from 5425 students in grades 3~5 of 12 primary schools in the urban area of Luzhou, China, we determined that more than two-thirds of students aged 7 to 14 years have experienced bullying in the past year. These data demonstrate an urgent need to seek the reasons for this behavior and to adopt, scale, and sustain evidence-based interventions to reduce this high burden of school bullying. We also found a positive relationship between primary school bullying responses and psychological resilience among victims, with differences by sex. Therefore, the findings indicate that we may be able to prevent and intervene in school bullying by increasing students’ level of psychological resilience. In addition, we should analyze and explore the contributing factors of school bullying and formulate effective school bullying interventions from multiple perspectives, especially sex, bullying roles, and psychological resilience, to reduce the frequency of school bullying incidents.

## Figures and Tables

**Figure 1 ijerph-19-16121-f001:**
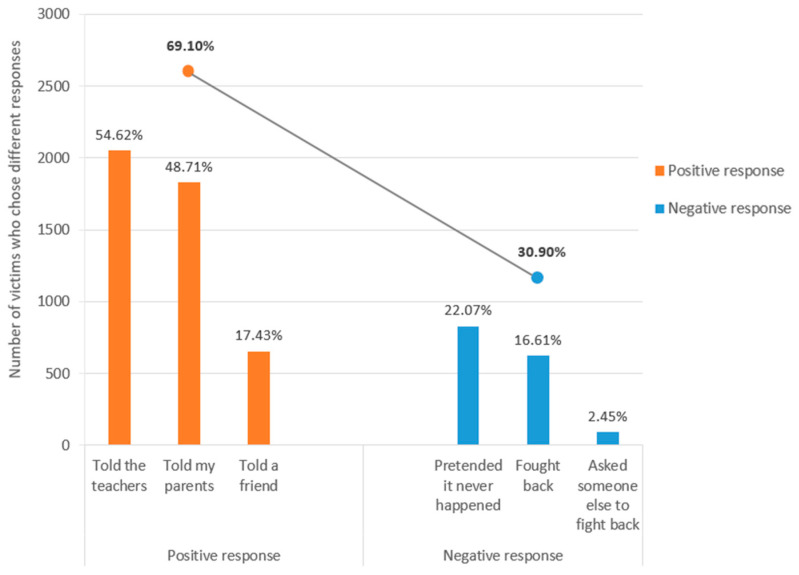
Victims’ responses to school bullying in primary schools of Luzhou City.

**Table 1 ijerph-19-16121-t001:** Sociodemographic characteristics of the participants in Luzhou City, China [*n*(%)/$(\bar{x}\pm s$)].

Sociodemographic Characteristics	$n(\%)/(\bar{x}\pm s)$	Sociodemographic Characteristics	$n(\%)/(\bar{x}\pm s)$
Enrolled school		Having bad health behavior ^#^	
Public	4379 (80.72)	Yes	3549 (67.06)
Private	1046 (19.28)	No	1743 (32.94)
Districts		Sufficient sleeping time ^#^	
Jiangyang	1886 (34.77)	Yes	3080 (57.29)
Longmatan	2134 (39.34)	No	2296 (42.71)
Naxi	1405 (25.90)	Father’s education level ^#^	
Sex ^#^		College and more	1486 (27.39)
Male	2792 (52.15)	High school/middle school	2561 (47.21)
Female	2562 (47.85)	Primary school or less	531 (9.79)
Grade		Unknown	847 (15.61)
3	1589 (29.29)	Mother’s education level ^#^	
4	1894 (34.91)	College and more	1410 (25.99)
5	1942 (35.80)	High school/middle school	2551 (47.02)
Character ^#^		Primary school or less	649 (11.96)
Quiet	725 (13.48)	Unknown	815 (15.02)
General	1989 (36.98)	Parents are the main family educator ^#^	
Outgoing	2664 (49.54)	Yes	4329 (81.30)
Number of good friends ^#^		No	996 (18.30)
≤1	299 (5.56)	Parents quarreled before children	
≥2	5079 (94.44)	Yes	1571 (29.39)
Class cadre participation ^#^		No	3775 (70.61)
Yes	1971 (36.72)	Divorced parents ^#^	
No	3397 (63.28)	Yes	659 (12.18)
Academic performance ^#^		No	4751 (87.82)
Above average	1827 (33.97)	The way of education ^#^	
Average	3023 (56.21)	Negative	848 (16.07)
Below average	528 (9.82)	Both	1431 (27.11)
Relationship with siblings ^#^		Positive	2999 (56.82)
Good	3297 (89.11)	Psychological resilience	
General	312 (8.43)	High	1905 (35.12)
Poor	91 (2.46)	Middle	1820 (33.55)
Age	9.56 ± 1.56	Low	1700 (31.34)
Score of recognition	63.57 ± 6.18	Psychological resilience level	40.52 ± 8.04

Notes. ^#^ missing values.

**Table 2 ijerph-19-16121-t002:** Response to school bulling among victims with different characteristics [*n*(%)/$(\bar{x}\pm s$)].

Variable	Positive Response	Variable	Positive Response
Districts **		Sufficient sleeping time **	
Jiangyang	821 (63.64)	Yes	1489 (72.53)
Longmatan	1074 (74.27) ^a^	No	1080 (64.86)
Naxi	701 (68.66) ^a,b^	Mother’s education level *	
Sex **		College and more	585 (64.78)
Male	1303 (64.96)	High school/middle school	1232 (70.04) ^a^
Female	1261 (74.18)	Primary school or less	358 (71.60)
Grade *		Unknown	421 (70.76)
3	847 (72.21)	Parents as the main family educator **	
4	904 (69.33)	Yes	2045 (70.01)
5	845 (66.02) ^a^	No	496 (65.01)
Character *		Parents quarreled before children	
Quiet	364 (72.51)	Yes	821 (61.96)
General	978 (65.64) ^a^	No	1732 (72.99)
Outgoing	1231 (71.16) ^b^	Divorced parents *	
Number of good friends *		Yes	330 (63.71)
≤1	152 (61.04)	No	2257 (69.94)
≥2	2418 (69.70)	Education method **	
Academic performance *		Negative	434 (61.47)
Above average	748 (65.79)	Both	691 (63.16)
Average	1537 (70.93) ^a^	Positive	1409 (75.59) ^a,b^
Below average	290 (69.21)	Age *	(9.48 ± 0.97)
Relationship with siblings **		Recognition score **	(63.25 ± 5.96)
Good	1607 (72.35)	Psychological resilience level **	
General	165 (64.02) ^a^	Low	972 (65.81)
Poor	42 (50.60) ^a^	Middle	870 (68.29)
Bad health behaviors **		High	754 (74.95) ^a,b^
Yes	1802 (66.01)		
No	721 (77.61)		

Notes. * *p* < 0.05, ** *p* < 0.001. The test level is *α* = 0.05; “a” indicates that the difference compared with the first group is statistically significant, “b” indicates that the difference compared with the second group is statistically significant.

**Table 3 ijerph-19-16121-t003:** Logistic regression analysis of factors influencing victims’ response to school bullying [*OR* (95% *CI*)].

Variables	Male	Female	Total
Psychological resilience level (ref: Low)			
Middle	2.046 (1.455~2.878) **	-	1.416 (1.108~1.810) **
High	2.300 (1.624~3.259) **	-	1.605 (1.254~2.055) **
Grade (ref: 3)			
4	-	-	0.707 (0.555~0.901) *
5	-	-	0.876 (0.703~1.090)
Female (ref: Male)	-	-	0.638 (0.530~0.768) **
Character (ref: Quiet)			
General	-	0.518 (0.317~0.847) *	-
Outgoing	-	0.933 (0.695~1.254)	-
Relationship with siblings (ref: Good)			
General	0.451 (0.242~0.838) *	0.430 (0.206~0.898) *	0.455 (0.279~0.743) *
Poor	0.547 (0.271~1.103)	0.620 (0.272~1.411)	0.589 (0.339~1.025)
Mother’s education level (ref: College and more)			
High school/middle school	2.297 (1.477~3.570) **	-	1.635 (1.191~2.246) *
Primary school or less	1.525 (1.031~2.256) *	-	1.126 (0.847~1.498)
Unknown	1.400 (0.875~2.242)	-	1.036 (0.733~1.463)
Parents are the main family educator (ref: No)	-	0.715 (0.519~0.986)*	0.827 (0.660~1.036)
The way of education (ref: Negative)			
Both	1.856 (1.352~2.547) **	1.810 (1.225~2.675) *	1.822 (1.423~2.333) **
Positive	1.612 (1.206~2.156) **	2.259 (1.664~3.067) **	1.857 (1.505~2.291) **
Sufficient sleeping time (ref: No)	0.689 (0.537~0.884) **	0.742 (0.564~0.977) *	0.747 (0.621~0.899) *

Note: All models were conducted by controlling the covariates: The method of education, parents are the main family educator, and sleeping time. The covariate sex was controlled only for model 3. * *p* < 0.05, ** *p* < 0.001.

## Data Availability

Due to privacy constraints, data are available upon request.

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
