# Peer review of "Psychological Resilience May Be Related to Students’ Responses to Victims of School Bullying: A Cross-Sectional Study of Chinese Grade 3–5 Primary School Students"

_ijerph, 2022, doi:10.3390/ijerph192316121_

Round 1

Reviewer 1 Report

The article deals with issues of great importance for many disciplines of modern science. The problem of children's reaction to school bullying has a universal dimension. The results of this type of research are of interest to researchers of psychology, public health, and the theory and practice of education. Developing effective methods of preventing the negative effects of the presence of a small child as a victim of bulling is a difficult task to perform without the support of interdisciplinary research results.

The authors accurately defined the subject and objectives of the research. The main advantage of this project is a well-chosen, random sample of 5,425 primary school students in China. I have no objections to the methodological assumptions adopted in the project, the method of conducting research and the methods of analyzing the collected empirical data. However, I suggest considering a few points.

1. The research project was of a diagnostic and verification nature. However, the main goal of the authors was to verify the thesis that there is a relationship between psychological resilience and the reaction of school bullying victims. For this purpose, a research hypothesis should be formulated and the procedures for its verification should be very clearly defined. Unfortunately, this point has been omitted. A hypothesis is a logical conclusion drawn from the existing theories and the results of research carried out so far. The authors cite many research reports in the article, but I do not find any references to psychological theories justifying the existence of the assumed dependencies

2. The method of presenting the obtained results is also unsatisfactory. I believe that the graphical representation of the research results (e.g. in the form of charts) can make the text more user-friendly. In the current editorial form, the reader may get lost in an "avalanche" of numbers.

3. It is also worth considering the questions of data interpretation contained in the part of the article entitled "discussion". Appealing to theories developed in the field of psychology and education could enhance the academic validity of this article.

Generally, however, the research presented in the article meets the criteria of reliability and validity - hence I recommend their publication.

Author Response

Responses to Reviewer 1 Comments

Manuscript ID: ijerph-1972913

Title: Psychological Resilience May Be Related to Students' Responses to Victims of School Bullying: A Cross-Sectional Study of Chinese Grade 3-5 Primary School Students

Dear Reviewer,

We are truly grateful for the comments and suggestions from you. We have made careful modifications to the original manuscript. We hope that the new manuscript will obtain your approval. Below you will find our point-by-point responses to your comments.

Comment 1: The research project was of a diagnostic and verification nature. However, the main goal of the authors was to verify the thesis that there is a relationship between psychological resilience and the reaction of school bullying victims. For this purpose, a research hypothesis should be formulated and the procedures for its verification should be very clearly defined. Unfortunately, this point has been omitted. A hypothesis is a logical conclusion drawn from the existing theories and the results of research carried out so far. The authors cite many research reports in the article, but I do not find any references to psychological theories justifying the existence of the assumed dependencies.

Response 1: The suggestion of finding theoretical support is indeed very constructive. Supplements about the theoretical model and the procedures for verification have been added to the background part of the article.

Changes in the text: background section, page 2, lines 80–106.

Comment 2:The method of presenting the obtained results is also unsatisfactory. I believe that the graphical representation of the research results (e.g. in the form of charts) can make the text more user-friendly. In the current editorial form, the reader may get lost in an "avalanche" of numbers.

Response 2: We have modified our manuscript.

Changes in the text: Results section, page 6, lines 243-245.

Comment 3: It is also worth considering the questions of data interpretation contained in the part of the article entitled "discussion". Appealing to theories developed in the field of psychology and education could enhance the academic validity of this article.

Response 3: Thank you for your valuable comments. The corresponding discussion part has been improved.

Changes in the text: Discussion section, page 9-10, lines 305–315,327-331,337-342,356-361.

Special thanks to you for your good comments.

Reviewer 2 Report

This study is meaningful in terms of providing the information of bullying in China which has not well known yet. However, the overall manuscript need to more clarified.  

Introduction

The needs of the study should be justified; resilience as a protective factor of impact of bullying should need mor explanation, currently it is only mentioned for line 81-84, p.2.

p.4., line 17. “Respondent selected at least one negative option in the answer to indicate that the answer was negative, otherwise it was positive” è It is not clear how the negative, positive answers were finally coded.

This study used Owleus questionnaire to ask students’ bullying and victim experiences’ however the results provided information only about prevalence of victim; it is curious prevalence of bullying (also there might be bully-victim).

p.4. line 19~ Statistical analysis part needs to be specified; the details of the process of analysis are needed for each aim (e.g. aim2: chi test for examining the relation between, t-test for examining the relation which variables? aim 3: predictor? Outcomes?)

Information of the tables need to more clarified; the end of title of the each table indicates “ (n(%)x)” , but the numbers of the table seem to indicate number and percentage, that is “n(%)”

Table 3 and 4 are difficult to understand. The information given is very confused. The way of providing information needs to be more consistent.  

p.6.line22~ the title “3.3 frequency distribution of the characteristics of victims’ does not fit the paragraph and table. The table 3 seems to explain frequency (number and percentage) of positive responses? The asterisk (*) indicate significance level; however, the differences more than 2 groups (e.g. grade, character, relationship with siblings etc…) are not clearly shown.

Similarly the title “3.4 relationship between psychological resilience and responses to school bullying by sex” does not fit the paragraph and table 4. It is quite difficult to follow the information and results.

This survey seems to have fruitful information but the result section was not appropriately organized.

Also, the focus of this study is ‘psychological resilience’, therefore the mechanism of students’ resilience and positive responses of bullying should be further analyzed with the authors’ interpretation.

Sex differences is suddenly shown from the result and discussion part, so, this information should be explained in introduction if the authors regard the sex is meaningful variable.

Hope the comments are helpful for revising the manuscript.

Author Response

Responses to Reviewer 2 Comments

Manuscript ID: ijerph-1972913

Title: Psychological Resilience May Be Related to Students' Responses to Victims of School Bullying: A Cross-Sectional Study of Chinese Grade 3-5 Primary School Students

Dear Reviewer,

We are truly grateful for the comments and suggestions from you and the other reviewers. We have made careful modifications to the original manuscript. We hope that the new manuscript will obtain your approval. Below you will find our point-by-point responses to the reviewers’ comments.

Comment 1:The needs of the study should be justified; resilience as a protective factor of impact of bullying should need more explanation, currently it is only mentioned for line 81-84,

Response 1: Thank you for your valuable comments. Supplements about the theoretical model and the procedures for verification have been added to the background part of the article.

Changes in the text: background section, page 2, lines 81–107.

Comment 2: p.4., line 17. “Respondent selected at least one negative option in the answer to indicate that the answer was negative, otherwise it was positive” è It is not clear how the negative, positive answers were finally coded.

Response 2: We have modified our manuscript.

Changes in the text: Method section, page 4, lines 189-194.

Comment 3: This study used Owleus questionnaire to ask students’ bullying and victim experiences’ however the results provided information only about prevalence of victim; it is curious prevalence of bullying (also there might be bully-victim).

Response 3: We have modified our manuscript.

Changes in the text: Results section, page 6, lines 239-241.

Comment 4: p.4. line 19~ Statistical analysis part needs to be specified; the details of the process of analysis are needed for each aim (e.g. aim2: chi test for examining the relation between, t-test for examining the relation which variables? aim 3: predictor? Outcomes?)

Response 4: We have modified our manuscript.

Changes in the text: method section, page 5, lines 208-222.

Comment 5: Information of the tables need to more clarified; the end of title of the each table indicates “ (n(%)x)” , but the numbers of the table seem to indicate number and percentage, that is “n(%)”

Response 5: After checking, we found that this may have been a misunderstanding. In Tables 1 and 2, although the vast majority are categorical variables, there are also age, Score of recognition, and psychological resilience, all three quantitative variables are described using means and standard deviations, hence the representation in the section of the table title.

Comment 6: Table 3 and 4 are difficult to understand. The information given is very confused. The way of providing information needs to be more consistent.p.6.line22~ the title “3.3 frequency distribution of the characteristics of victims’ does not fit the paragraph and table. The table 3 seems to explain frequency (number and percentage) of positive responses? The asterisk (*) indicate significance level; however, the differences more than 2 groups (e.g. grade, character, relationship with siblings etc…) are not clearly shown.

Response 6: We have modified our manuscript. After thinking, On the basis of references to other similar literature [1], we set the title is “Response to School Bulling among Victims with Different Characteristics”.

Changes in the text: Results section, page7-8, lines 255-275.

[1].Peng, Y., et al., Status and Determinants of Symptoms of Anxiety and Depression among Food Delivery Drivers in Shanghai, China. Int J Environ Res Public Health, 2022. 19(20).

Comment 7: Similarly, the title “3.4 relationship between psychological resilience and responses to school bullying by sex” does not fit the paragraph and table 4. It is quite difficult to follow the information and results.

Response 7: We have modified our manuscript. After weighing and thinking, based on the reference to other similar literature[1], we set the title is “Logistic Regression Analysis of Factors Influencing Victims' Response in School Bullying”.

Changes in the text: Results section, page 7, lines 260; page 8, lines 271-272.

[1].Peng, Y., et al., Status and Determinants of Symptoms of Anxiety and Depression among Food Delivery Drivers in Shanghai, China. Int J Environ Res Public Health, 2022. 19(20).

Comment 8: This survey seems to have fruitful information but the result section was not appropriately organized.

Response 8: According to your valuable comments, we have made some changes to the headings and table headings in the results section of the manuscript.

Changes in the text: Results section, page 5-8, lines 228-273.

Comment 9: Also, the focus of this study is ‘psychological resilience’, therefore the mechanism of students’ resilience and positive responses of bullying should be further analyzed with the authors’ interpretation.

Response 9: Thank you for your valuable comments. The corresponding discussion part has been improved.

Changes in the text: Discussion section, page 9-10, lines 305–315,327-331,337-342,356-361.

Comment 10: Sex differences is suddenly shown from the result and discussion part, so, this information should be explained in introduction if the authors regard the sex is meaningful variable.

Response 10: In the background section, we have added this information to our manuscript.

Changes in the text: Discussion section, page 3, lines 117–120.

Special thanks to you for your good comments.

Round 2

Reviewer 2 Report

The manuscript was well revised by the comments